# Lessons Learned with a Triad of Stakeholder Advisory Boards: Working with Adolescents, Mothers, and Clinicians to Design the TRUST Study

**DOI:** 10.3390/children10030483

**Published:** 2023-03-01

**Authors:** Alexis Richards, Marissa Raymond-Flesch, Shana D. Hughes, Yinglan Zhou, Kimberly A. Koester

**Affiliations:** 1Department of Pediatrics, Division of Adolescent and Young Adult Medicine, University of California San Francisco, San Francisco, CA 94107, USA; 2Department of Pediatrics, Division of Adolescent and Young Adult Medicine, Philip R. Lee Institute for Health Policy Studies, University of California San Francisco, San Francisco, CA 94107, USA; 3Vitalant Research Institute and Department of Laboratory Medicine, University of California San Francisco, San Francisco, CA 94105, USA; 4Department of Neurology, Memory and Aging Center, University of California San Francisco, San Francisco, CA 94158, USA; 5Department of Medicine, Division of Prevention Science, Center for AIDS Prevention Studies, University of California San Francisco, San Francisco, CA 94158, USA

**Keywords:** advisory boards, youth, mothers, sexual and reproductive health, alone-time, adolescents, parents, triadic engagement, technology

## Abstract

Optimal care for pediatric and adolescent patients is carried out under a triadic engagement model, whereby the patient, caregiver, and clinician work in collaboration. Seeking input from all triad members in the development and implementation of clinical trials and interventions may improve health outcomes for children and adolescents. Sufficient evidence demonstrating how to effectively engage stakeholders from all branches of this triadic model is lacking. We address this gap by describing the successes and challenges our team has encountered while convening advisory groups with adolescent patients, parent stakeholders, and their clinicians to assist in the development and deployment of a technology-based intervention to promote the utilization of sexual and reproductive health services by increasing adolescent–clinician alone-time. Each stakeholder group contributed in unique and complementary ways. Working with advisors, our team aligned the priorities of each group with the goals of the research team. The results were improvements made in the content, design, and delivery of the TRUST intervention. While we were largely successful in the recruitment and engagement of adolescent patients and clinicians, we had less success with parents. Future research will need to explore additional strategies for recruitment and engagement of parents, particularly in rural, minority, and underserved communities.

## 1. Introduction

There has been a global movement towards engagement of key stakeholders, including patients, caregivers, and clinicians in the research process. In pediatric research, involvement of stakeholders in research can impact adolescent and pediatric outcomes, as well as improve the engagement of people from underserved communities [1,2,3,4,5]. Community advisory boards (CABs) are a useful mechanism to collaborate with stakeholders. CABs have been used to strengthen academic–community partnerships and promote the needs of underserved communities in clinical and research settings [3,5,6,7]. Meaningful use of advisory boards can lead to collaborative development of research questions, approaches to recruitment and data, as well as guide interpretation and study dissemination [2,4,5,7]. Given their intended purpose and applications, CABs may be an appropriate vehicle for facilitating research engagement with the triad of stakeholders critical in adolescent healthcare: adolescents, parents, and clinicians [8,9,10]. The interactions between the different arms of the triad are shown in Figure 1.

Adolescence is a period of rapid and varied physical, social, and cognitive development [11,12], which requires tailored approaches to appreciably engage youth in research. To achieve this, some researchers have employed youth advisors (YAs); however, youths remain underrepresented in clinical research [13]. Engaging YAs in the development and implementation of clinical trials has a bidirectional impact: research teams gain relevant insights through the YA perspective and lived experiences, and the YAs gain experience participating in projects to improve health, leading to empowerment and sense of inclusion [13,14,15,16,17,18]. Published literature on methods of engaging YAs is lacking, leading to a gap in how to effectively partner with YAs in the research process [1].

Clinicians are also key stakeholders in research that impacts both individual- and population-level patient care, as they bridge the gap between communities where care is delivered and the academic centers where new interventions are developed [19,20,21]. Involving representatives from community clinics and clinicians in research advisory boards can help to develop and test interventions, improve translation of novel interventions into clinical practice, and assist with dissemination [20,21,22]. This is consistent with Forysthe et al.’s findings that clinicians are interested in active engagement in clinical research with the goal of improving patient care and contributing to the advancement of evidence-based practice [21].

Parents, and particularly mothers, play a critical role as gatekeepers in the care of their children. Urkin et al. conducted a prospective study investigating who accompanies children and adolescents to their medical visits and found that 90% were accompanied by a parent, the majority of which were mothers [23]. The use and impact of parent or mother advisory groups has not been well documented in the pediatric literature; however, there has been significant research supporting the involvement of caregivers in medical decision-making to improve patient outcomes [9,10,24,25,26,27]. While there is still a scarcity of published data, the use of parents in the development of pediatric interventions has been increasing in the last decade, as parent involvement can improve pediatric research engagement by increasing participant enrollment and retention, contributing to the development of interventions and measurement tools, and identifying potential issues in a research project [19,24,28,29,30,31].

Seeking input from all triad members in the development and implementation of clinical trials and interventions may improve health outcomes for children and adolescents. However, we lack models for effectively engaging all branches of the adolescent healthcare triad as research advisors. This may be because engagement of adolescents, caregivers, and clinicians can be especially complex in pediatric research for three reasons. First, an adolescent’s legal rights to confidential care in some areas (e.g., reproductive and sexual health) can complicate engagement between parents and clinicians. Second, adolescent patients steadily increase in their capacity for medical decision-making and bodily autonomy, so the nature of their participation should reflect their capabilities, while also respecting that the parent retains legal responsibility for a child’s welfare [29]. Considerations of confidentiality notwithstanding, adolescent participation in research may confront the need for parental consent in addition to youth assent. While parental consent is mandatory for research participation of minors, youth assent can increase patient engagement and investment into the research process through increasing youth autonomy and comprehension [15,32]. Finally, adolescents may not feel comfortable voicing opinions and concerns when in the company of clinicians and/or parents, which may limit their participation in clinical decision-making and research [18,33,34].

Utilization of sexual and reproductive health services among adolescents in the United States is unacceptably low, especially given the higher burden of negative sexual and reproductive outcomes in youth aged 15–19 years and those in the Latinx and Black communities [8]. Alone-time with primary care clinicians improves health outcomes in adolescents by improving access to preventative services, including sexual and reproductive health services [35,36,37,38,39]. Parents, particularly mothers, are instrumental in increasing adolescent utilization of confidential sexual and reproductive health services, as well as increasing effective communication between adolescents and their clinicians [4,9,10,22,31,40,41,42]. Despite studies showing that most parents believe alone-time is important, a recent analysis showed that only 40% of adolescents had alone-time with their clinician at their last visit [43], and adolescents from families with lower income, without health insurance, and of Latinx ethnicity are even less likely to have alone-time with their clinicians [43]. Under ideal conditions, alone-time is the outcome of efforts among the triad of the patient, parents, and clinicians all working together to encourage teens to become self-advocates and independently navigate the healthcare system.

In this paper, we describe our efforts to promote these ideal conditions by drawing on lessons learned as we worked to engage the triad of adolescent, parent, and clinic partner advisory boards in the design of a novel intervention to promote adolescent–provider alone-time.

## 2. Materials and Methods

### 2.1. TRUST Intervention

The Technology-based Resources to Increase Uptake of Sexual Health Services for Teens (TRUST) study, which began in September 2020, aims to improve communication between teens, parents, and clinicians about sexual and reproductive health by increasing the frequency of alone-time between adolescents and their clinicians. The TRUST study consists of interactive modules, developed during the first phase of the study for racially and ethnically diverse youth (aged 11–17 years) and their mothers, and designed to be accessible and usable by all communities. Technology-based interventions have proven to be effective in promoting healthy behaviors and can be accessed easily by most communities [44,45,46]. TRUST is hosted on a youth-oriented health and wellness website (GritX.org) and focuses on four topics: (1) The Adolescent Well Visit: Check Up & Check In; (2) General Communication; (3) Talking about Relationships and Sexual Health; and (4) Parental Monitoring. A description of the modules is listed in Table 1.

The TRUST Study is currently in the feasibility phase and is being conducted in accordance with the Declaration of Helsinki, and is approved by the Institutional Review Board of University of California, San Francisco (Study Number 20-32013).

### 2.2. Clinical Settings

Our research team partnered with three primary care clinic sites to develop and deploy the TRUST intervention. The first site is an urban primary care and specialty care clinic staffed by adolescent medicine specialists in San Francisco, serving patients 12–26 years of age, with a racially and ethnically diverse mixture of publicly and privately insured patients. The second site is a consortium of eight private pediatric and family medicine practices affiliated with a large non-profit community hospital in an urban Chinatown neighborhood. Together, these serve about 500 adolescents annually, of whom more than 90% are Chinese and Chinese Americans, with a majority that qualify for public insurance. The third site is a large academically-affiliated, multisite, community-based, federally qualified health center with a large rural catchment area, where approximately 72% of patients are Latinx. Our team chose these sites to allow sampling from different racial, ethnic, and rural/urban populations.

### 2.3. Advisory Board Engagement

In developing and refining the TRUST modules, our goal was to generate content that was realistic, thought-provoking, and engaging, with illustrations inclusive of diverse adolescents and parents. To achieve this, we sought iterative input from our advisory boards. Detailed recruitment and engagement strategies for each of the advisory boards can be found in Table 2.

#### 2.3.1. Clinic Partner Advisory Board

For our clinical advisory board, we solicited the participation of clinicians (healthcare providers and registered nurses) and clinic administration from each of our three collaborating clinical sites. The research team planned to meet with the clinic partner advisory board (CPAB) regularly through video conferencing and email to elicit input on findings of the literature review, translation of the findings into tailored content for the modules, mode of delivery, pilot study structure, and review preliminary results. We also planned to engage the CPAB to advise the team on recruitment strategies for participants in the youth and mother advisory boards, and participants for the feasibility study. Inclusion criteria included working at one of our clinical partner sites and having experience taking care of adolescents and their parents.

#### 2.3.2. Youth Advisory Board

Our research team sought to recruit racially and ethnically diverse adolescents living in California from the communities served by our clinical sites to serve on a youth advisory board (YAB). The purpose of the youth advisors was to provide feedback on the study design, assist in the interpretation of the literature review into tailored content from the lens of an adolescent patient, and pilot-test the modules for both functionality and content. In the following phase of the study, the research team planned to hold asynchronous focus groups and in-depth interviews to examine user experience of the modules.

Our research team recruited youth advisors through a Qualtrics survey distributed through several channels, including a mentoring program for minority students interested in medicine, presentations at a summer camp for students interested in medicine, recruitment flyers in our participating clinics, and via participant-driven referrals. The research team requested an 18-month commitment from the youth to share opinions on the study, help recruiting other teens for the advisory board, and help refining module content for use in the pilot study. Inclusion criteria included being an adolescent between 13–17 years, having stable access to the internet, and having a phone that can receive text messages. There were no formal exclusion criteria. Each youth needed a signed parental consent to participate as an advisor.

#### 2.3.3. Parent Advisory Board

The research team planned to assemble a parent advisory board (PAB) of mothers of patients from the three collaborating clinical sites. We planned to seek feedback regarding the content of the modules in parallel to the youth advisors, specifically focusing on parental monitoring, communication, and increasing alone-time between adolescents and clinicians. The research team aimed to engage the mothers in in-person focus groups to evaluate the content and functionality of the modules.

We sought to recruit two mothers from each of our partnering clinical sites to serve on our parent advisory board. The research team posted flyers in English, Spanish, and Chinese in our partnering clinical sites. The study team asked our youth advisors if their mothers would be interested in participating. Outreach was also performed through parenting groups on social media and through engagement of parents of teens outside of our target clinics, due to limited participation. Inclusion criteria included that participants must be mothers of adolescents between 13–17 years, have stable access to the internet, and have a cellphone that can receive messages. There were no formal exclusion criteria.

## 3. Results

Here, we describe the outcomes of the study team’s advisory board recruitment efforts, as well as the contributions of each of the advisory boards during the different phases of the TRUST study development (see Table 2).

### 3.1. Clinic Partner Advisory Board

#### 3.1.1. Recruitment Strategies

During the project development phase, the research team established a relationship with a representative at each of the clinic sites. Each site provided us with a letter of support. After the TRUST study received funding, we asked each partner site representative to assist in identifying and inviting individuals with relevant clinical experience caring for adolescents, and/or those in positions of leadership within their clinic to serve on the CPAB. We succeeded in recruiting seven professionals in total, with two or three from each individual site, including two clinic administrators, four medical doctors, and one registered nurse.

#### 3.1.2. Engagement Strategies

Communication occurred primarily over secure email and video conferences. Email communication was used for asynchronous feedback and meeting coordination. The CPAB met quarterly with the research team via Zoom teleconference calls at times agreed upon by a quorum of the CPAB. The meetings occurred either during the lunch hour or at the end of the workday to accommodate busy clinical schedules.

#### 3.1.3. Major Contributions

##### Module Development and Refinement

The research team utilized the CPAB during the initial development of the TRUST online modules, refinement of module content, and in the recruitment of participants for the youth advisory board, parent advisory board and feasibility study. During the module development phase, we sought feedback via the engagement strategies described above from our clinic partners on module content and themes that they viewed as important for promoting adolescent-provider alone-time. After developing the initial content for each module (see Table 1 for module descriptions), the CPAB provided feedback on the language, messaging, and module appropriateness for parent and youth audiences. They also provided guidance on literacy levels, as well as the predominant languages spoken in their communities.

##### Recruitment of Advisory Boards and Recruitment of Participants for the TRUST Study

As each clinic served a unique geographic and racial/ethnic population, the clinic advisory board also assisted us in directing our recruitment strategies for our youth and parental Advisory Boards, as well as for participant recruitment in our feasibility study. Recruitment approaches varied by target audience, site and phase of the project and ranged from the placement of flyers in clinical sites to the use of patient portals in electronic health record systems to recruit patients for the feasibility phase of the TRUST study (see below for additional details by population recruited).

#### 3.1.4. Challenges

The research team continuously worked on learning how best to engage the CPAB through the different phases of the TRUST study. Due to the COVID-19 pandemic, increased clinical demands and busy schedules offered limited availability for quarterly meetings of CPAB. One strategy we deployed to address this challenge included sending CPAB members an electronic gift certificates to be used to purchase lunch. In addition to the COVID-19 pandemic limiting in-person engagement, as one of the clinical sites is also a significant geographical distance from the research team, it was difficult for the team to travel to the site to troubleshoot any difficulties with participant recruitment and engagement.

### 3.2. Youth Advisory Board

#### 3.2.1. Recruitment Strategies

Thirty-nine youths responded to our recruitment efforts. We invited 14 to participate, selecting for varied ages, gender identities, and race/ethnic backgrounds. Four youths were no longer interested, two youths did not respond, and 8 youths assented to join the YAB. Our YAB is 12% Black, 62% Asian, and 25% White; 75% female-identifying and 25% male-identifying; and 87.5% cisgender and 12.5% transgender.

#### 3.2.2. Engagement Strategies

Youth were primarily contacted through email or texting for the initial parental consent and youth assent to participate in the youth advisory board. YAs were then asynchronously assigned advisory tasks through email. Using a dedicated cellphone, the research team also used text messages to send reminders and updates to the youth advisors. There were one or two youth advisors that did not have cellphones but were able to communicate with the research team through email.

#### 3.2.3. Major Contributions

##### Module Development

Using Google Docs, our YAs reviewed and provided feedback on module content. We asked each of our youth advisors to complete two review tasks during the development of the modules and provided $20 per completed task. We asked the youths to relay the key messages and takeaways from each module in their own words, rephrase the messages to highlight content they viewed as important, list content that they felt was missing, and comment on the readability of the modules. Feedback was interactive and collaborative, with YAs making comments and reading and responding to peer comments. We also asked the YAs to comment on which illustrations they preferred or with which they felt a connection. The research team also asked the YAs if they thought any content was missing from the modules that they thought was salient and should be included.

##### Module Refinement

In the next phase of the study, we asked our YAs to pilot-test the online interactive modules through Recollective software, a qualitative research platform. The platform enabled the YAs to make visual and audio recordings as they moved through the modules online in real-time. We asked the youths for real-time feedback as they were going through the modules, focusing on content appropriateness, youth-tailored language, usability/technical challenges, repetitiveness, and/or questions that were raised during the work-through. The research team then reviewed these recordings for module content refinement and to problem-solve technical challenges. YAs conducted this pilot testing individually and, in some cases, we requested pairs of YAs view content together and discuss what they liked and did not like. Seven YAs completed individual reviews of one or two modules, and three pairs reviewed the Well Adolescent Visit: Check Up and Check In module together in the Recollective software. YAs were compensated $25 per module reviewed.

When asked their thoughts about the overall importance of the TRUST modules the youths highlighted that many of the topics in the modules were not always discussed at home or at school: “My initial reaction to the concept of the TRUST intervention was a really positive one, because I know a lot of peers who struggle to discuss these personal matters with their trust[ed] adults and I feel like this intervention serves as a good tool to break the ice and actually start these conversations.” Others were surprised to find out that many adolescents do not have alone-time with their clinicians: “I was curious since I never knew [that having alone-time with a provider] was an issue. I think if more people knew about it, there would be a lot more emphasis to get children to bond with their parents and healthcare providers.”

After working through the modules, we asked the YAs to provide feedback on the most salient lessons learned. One youth commented: “I thought the information was really informative. The most surprising fact is that only 10% of teens get [well] checks. I think that most would be interested in learning the information in this module.” The key take aways were generally aligned with the goals of the research team: “1. I think the content was good and interesting and engaging. 2. I learned that I should talk to my healthcare provider alone more. 3. I think peers would be interested in reacting to this [module] because they would gain a lot of knowledge.”

We asked our youth advisors to provide feedback on the functionality of the modules, including the flow, design, images, and layout. One youth commented: “(1) It was aesthetically pleasing. (2) The drawings were fun and made the [module] more comfortable. (3) It was difficult having to scroll down to read the passages then scroll back up to answer (4) I pushed through, but took a small break. I like the idea of it being spaced out.” Another reported: “I like the idea of breaks, especially [if] it is not mandatory, so we can decide to take it or not. I think the font of the lesson could be a size bigger. I liked the text message type of layout of the pages. The only thing I think could be changed is the placement of the images.”

Finally, we asked our YAs to comment on module content that they thought was missing from the TRUST interventions. We identified several content themes that the YAs would have liked to see, including addressing LGBTQ identities and mental health. One YA commented: “I think it is important to talk about self-identity. If someone is exploring their identity, whether it comes to sexuality or gender, they should be able to talk to their parents about it without feeling judged.” Another commented: “Having close friends who are part of this community makes me wish it was normalized to be talked about more when I was younger. It would have given me more information on how to be an ally.” The YAs also identified mental health and stress as being something that many of them and their peers struggle with, leading to a need for more information and skills to talk about it with their parents. One YA commented: “I think anxiety, depression, and mental health in general is pretty important for all teens to talk about with their parents. I have struggled with it, so for me I find it important to discuss it with my parents.” Another commented: “I think mental health issues are big amongst teens and a big part of it gets shut out from parents. I think talking about mental health and creating a safe space to talk about it between parents and their teens is important.”

#### 3.2.4. Challenges

Due to the COVID-19 pandemic, the research team was limited to virtual interactions during our advisor recruitment, module development, and module refinement. While the research team had envisioned most of the participation from the YAs to be virtual, the necessity for all recruitment and engagement virtual was limiting. In terms of engagement from the participating YAs, all our youth advisors were students, and academic demands sometimes took priority over study-related tasks. The research team was successful in recruiting YAB members from urban settings, including youth of diverse gender identities, ages, and those from African-American, Asian, and White communities. We were unsuccessful in recruiting YAs from the from the predominantly rural Latinx community near one of our clinic sites.

### 3.3. Parent Advisory Board

#### 3.3.1. Recruitment Strategies

Two mothers responded to our recruitment efforts and consented to join the PAB. Additionally, we recruited a Bay Area sexual health educator and parent as an expert with extensive experience engaging the region’s parents and youth in discussing reproductive health. This expert assisted our team with the development of module content during the initial phase of the study, then provided her perspective as parent and parent educator to inform later phases of the study.

#### 3.3.2. Engagement Strategies

Parents were contacted primarily using email or telephone for initial consent to participate in the advisory board. The research team also used ongoing email contact for asynchronous feedback and to coordinate bimonthly meetings during initial module development. Following the onset of the pandemic, when it was no longer safe or feasible to hold in-person focus groups, the research team attempted to hold asynchronous focus groups with PAB members, using the same online platform used with YAB members.

#### 3.3.3. Major Contributions

##### Module Refinement

We asked our parent advisors to give us feedback on the TRUST modules. Two of our parents reported that they thought the information in the modules was important for adolescents, parents, and clinicians. One stated: “[I] think it’s a wonderful concept and important to discuss.” Another said: “The purpose of the TRUST intervention seems like a worthy cause.” When asked what they thought was missing from the modules, one parent advisor commented: “Social dynamics and friendships. I’m not sure if this is already included, but self-esteem is a huge issue for teens.”

After consenting, the mothers, the parent advisors, including the sexual health educator, were provided a login and password to the Recollective Software, and asked to complete review of one module and provide real-time feedback through audio and video recording. While two of the mothers were able to give written feedback about the modules, none were able to use the online platform to capture real-time feedback. The sexual health educator contributed significantly to the salient content of our modules focused on sexual and reproductive health through regular video conferences and email exchanges.

#### 3.3.4. Challenges

The research team experienced significant difficulties in successfully engaging the parent advisors during module development. Due to the COVID-19 pandemic, the team was limited to virtual communications rather than in-person engagement. While we were able to elicit written feedback on the modules, we were unsuccessful in effectively guiding the participants from the Parent Advisory Group to record their pilot testing of the online modules through the online platform.

## 4. Discussion

Provider, patient, and caregiver engagement in advisory boards can make significant contributions to research design and processes [2,7,25,28,29,47,48]. However, there has been a paucity of studies published on the use of multiple and distinct advisory boards in research. Our research team was committed to including three different stakeholder groups to optimize the effectiveness of the TRUST intervention and ensure representation of all stakeholders in the study’s development, particularly adolescent voices. Overall, we were able to engage between 2–8 representatives within each group, making up the triad of stakeholders critical in adolescent healthcare, independently and in parallel throughout multiple phases in the TRUST study. Our research team sought to combine the engagement of frontline clinicians who interface with adolescent patients, with input from mothers and adolescents who are the intervention’s target audiences, to improve both the design and delivery of our intervention. Using our advisory boards, the research team was able to integrate the rich stakeholder feedback to adapt the content, including re-wording of key messages for parents or adolescents, adjusting images and content to ensure cultural and developmental appropriateness, and improving readability of the module content.

We used separate advisory boards, rather than one combined advisory board, to prevent inadvertent silencing of any stakeholders due to perceived power differentials and to allow for rich and open feedback from both our youth advisors and our parent advisors. Youth participatory research has shown that dilemmas can arise when balancing the perspectives of youth participants with other stakeholders, particularly in settings with established hierarchies—for example, clinical settings and schools [18,33,34]. In the clinical context, adolescents may be less likely to speak up and engage with clinicians when their parents are present to speak for them; it is possible that these patterns would translate to advisory board engagement in a mixed stakeholder group, making it more difficult to draw out youth perspectives [34]. Prior studies have shown that, in advisory boards that combine different groups of stakeholders, those with less experience in research, and perceived to be in lower positions of power, may not feel comfortable participating or sharing feedback [49,50].

While the research team was largely successful in the recruitment and engagement of our CPAB and YAB, we were less successful in both recruitment and engagement of mothers. This suggests the need for an early involvement with parent advisors to identify engagement approaches that are manageable against the competing demands for their time. As researchers, we failed to ask parents: how should we incorporate the perspectives of parents while developing our study? How do you want us to reach you? What can we realistically expect you to do? How can we make this easiest for you? Engaging caregivers is vital to improving healthcare engagement, patient outcomes, and satisfaction in an array of clinical settings and while data are sparse on the roles of parents in pediatric research, they likely play similar gatekeeper and facilitator roles as they do in the clinical context [9,10,24,25,26,27,31]. Parents may have barriers that either prevent or make them hesitant to engage in clinical research, such as obligations to their employment, considerable childcare, and household duties [32]. This suggests that different kinds of stakeholders will require different engagement strategies to overcome their unique barriers to participation and methods to address those barriers effectively.

Across different communities, stakeholder needs might also vary. For example, language and socioeconomic differences might necessitate offering alternative ways to share opinions or engage in research advising [32,51,52,53,54]. Compounding the differences in communities, Black and Latinx populations have historically been underrepresented in clinical research [48,54,55,56,57]. This underrepresentation may be due to a variety of reasons, including but not limited to a mistrust of the clinicians and academia, speaking little or no English, lack of access to care, lack of access to technology, lack of experience in research, transportation barriers, and having competing priorities, such as food or housing insecurity [48,54,55,56,57]. The COVID-19 pandemic worsened already existing structural racism and health disparities in social determinants of health experienced by communities of color in the United States—for example, increased morbidity and mortality from chronic disease, rates of unemployment, and decreased access to healthcare [58,59,60,61,62]. The disparities exacerbated by the pandemic [62,63,64] further limited the bandwidth of community members and clinicians for engagement in research. In the TRUST study, the research team’s interests and priorities were aligned with our stakeholders at the time we designed our study; however, the pandemic created a shift in resources and priorities, particularly for our partners in rural and historically marginalized communities. While we made every effort to ease research engagement for these stakeholders, we also had to accept that our study was not a priority for some of our advisors in the context of the pandemic.

In TRUST, we failed in several ways to forge a productive research collaboration with our newest clinical partner in an agricultural community serving primarily Latinx patients. Firstly, our research team was unable to travel in person to our rural community partner site to lead recruitment efforts. Secondly, during the pandemic, there was a heavy clinical burden placed on our partner sites, as noted, leading to limited resources for the clinic staff to participate in ongoing research projects. Lastly, in addition to the clinical strains, our rural site did not have Epic MyChart Access, which was used in other partner sites to connect with potential participants. As a result, we were unable to effectively recruit and engage parent and youth advisors at this site. Similar difficulties have been reported in other studies investigating recruitment and engagement of minority populations into research [62]. This suggests that, when research teams plan to engage research participants from historically marginalized or under-resourced communities, consideration should be given to competing demands for their time and resources, with an emphasis on exploring research that is of the greatest importance and urgency to the stakeholders.

In designing the TRUST intervention our team hoped that reliance on technology would allow for broad stakeholder engagement. As described, our research team had to pivot from hybrid to exclusively virtual recruitment and engagement of our advisory boards; we successfully used Recollective Software to collect feedback from our youth advisors. Generally, the use of a virtual platform creates the potential for a larger catchment area for advisors, with the possibility of greater involvement due to alleviation of travel burdens [8,26]. When designing our study, the research team had two competing hypotheses: virtual engagement will ease the access and engagement of more rural communities that would have been challenging to engage in-person; or rural communities will have decreased access to technology, leading to difficulty in both recruitment and participation. We found that, while virtual platforms facilitated retention and engagement of youth advisors, they were less effective for engagement of parents and in populations with lower technology penetration, consistent with other prior findings [8].

While this study adds to the limited literature about engaging the triad of adolescent, parent, and clinician advisory groups for the development of health interventions for adolescents, it has several limitations. Although we engaged clinical partners and youth from diverse communities, this study includes a small number of advisors, and whether the findings are transferable to more distinct geographic settings and populations is unknown. We were not successful in engaging parent advisors as well as youth advisors from one of our clinical sites, although we believe that sharing both our successes and challenges is important for advancing parent and youth advisory board engagement. Finally, our team recognizes that stakeholder engagement is an iterative and ongoing process, and we anticipate continuing to learn from our advisors as the TRUST Study progresses.

## 5. Conclusions

Research that aims to improve health outcomes in specific populations must involve the key stakeholders of that population. When designing modules to improve the uptake of confidential sexual and reproductive health services in adolescents, it was imperative to get the input of the adolescents themselves, their parents, and the clinicians who provide these sexual health and wellbeing services. Further research using multiple advisory boards is essential to develop best practices in the different recruitment and engagement strategies needed to integrate more isolated communities.

## Figures and Tables

**Figure 1 children-10-00483-f001:**
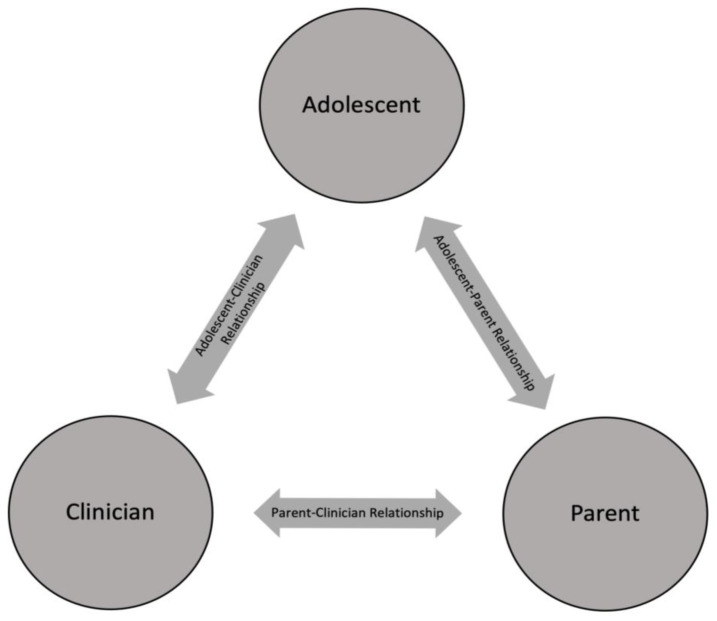
Triad of stakeholders in adolescent healthcare.

**Table 1 children-10-00483-t001:** TRUST modules and module descriptions.

Module Name	Module Description
The Adolescent Well Visit: Check Up & Check In	Overview of the importance of a Well Adolescent visit, how adolescents and their parents can plan for a well visit, and the introduction of alone-time between adolescents and their healthcare clinicians
General Communication	Information and tips to improve open and honest communication between adolescents and parents and skills to help when this communication is challenging
Talking about Relationships and Sexual Health	Information and skills to help parents and adolescents get on the same page about sexual health basics, make it easier to start a conversation on these topics, and each communicate effectively, with care and respect
Parental Monitoring	Information and skills on how to effectively monitor adolescents to ensure safety, improve health and wellness of the adolescent-parent relationship, and maintain appropriate boundaries and autonomy for the teen

**Table 2 children-10-00483-t002:** Advisory board recruitment, engagement, contributions, and challenges.

	Clinic Partner Advisory Board	Youth Advisory Board	Parent Advisory Board
Recruitment Strategies	Recruitment via: Research team’s professional network Primary study contact person at each clinic identified and invited appropriate representatives	Recruitment via: Qualtrics survey distributed through program for students interested in medicine Presentations given by research staff to students in summer camp program Recruitment flyers posted in study clinical sites Participant-driven referrals	Recruitment via: Recruitment flyers posted in study clinical sites Participant-driven referrals Recruitment of informal advisor who assisted research team with initial module development
Engagement Strategies	Email: Ongoing email contact for asynchronous feedback and meeting coordination Doodle polls sent to clinic advisors to identify optimal days and times to meet Quarterly meetings: Quarterly video conference meetings allowed for interactive discussions among clinicians from different clinical sites Occurred during lunch hour or at the end of the day to accommodate clinical schedules	Email: Youth provided email and/or cellphone contact information Youths were primarily contacted via email for task assignment Module development: Youth were emailed a set of questions along with the link to the Google Docs to guide their feedback Google Docs allowed advisors to read and comment on module content while interacting with comments provided by other advisors Module refinement: The research team used Recollective Software for individual or dyad feedback Recollective allowed for recorded audio and visual testing of the modules that could be performed asynchronously and then reviewed by the research team	Email: Ongoing email contact for asynchronous feedback and meeting coordination Weekly or biweekly text reminders sent by project coordinator Quarterly meetings: Biweekly video conference meetings for 1–2 months with the sexual health educator and the research team allowed for collaborative feedback and rich discussion
Major Contributions	Module development: Explored barriers and facilitators to increasing utilization of sexual and reproductive health services Suggested methods to increase alone-time between clinicians and teens Suggested methods to increase parent/adolescent communication Module refinement: Improved readability and appropriateness of content Gave language translation recommendations for each clinical site Advised on recruitment of mother and youth advisors: Advised on successful YA and parent advisory board recruitment approaches Assisted with placement of recruitment flyers in clinical sites or distribution to potential participants Recruitment of mother–teen dyads for feasibility study: Advised on strategies to recruit participants for feasibility study at each clinical site, including generating simple and concise recruitment language Placed and distributed of recruitment flyers Recruited eligible patients	Module development: Provided feedback on the key message and takeaways from initial module text Identified content that was missing from each module that they believed was important Module refinement: Reviewed illustrations and provided feedback on how relatable and inclusive they were Recorded pilot-testing of modules for real-time feedback regarding content appropriateness and youth-tailored language, usability, repetition, or questions from a youth perspective	Module development: Informal advisor contributed significantly to the salient content of modules Module refinement: Gave feedback on importance of TRUST interventions from parent perspective Gave feedback on topics that they thought were important but missing from the TRUST modules
Challenges	Quarterly meetings: COVID-19 demands on clinical practices created limited windows of time for clinical staff to meet Recruitment of advisors: One of the clinical sites was a significant distance from research team, limiting ability of the research team to travel to the site Each clinical site had different experience with research that posed barriers or facilitators to recruitment of advisors Some clinical sites had leadership that was hesitant regarding involvement in research Recruitment of participants for feasibility study: Only some clinical sites used a patient portal for messaging, limiting ability to contact patients for recruitment at some sites Clinical site leadership hesitation to participation in research	COVID-19 challenges: The research team was limited to virtual communication and meetings with youth advisors instead of in-person meetings Recruitment of youth advisory group: The research team faced significant challenges in recruiting youth advisors from rural areas The research team had difficulty in recruiting Latinx or Black advisors in our clinical settings Module development and refinement: Students had a variable schedule for other obligations that limited ability to engage at different stages of the study Some youth advisors did not have their own cell phone so could not be reached directly other than via email	COVID-19 challenges The research team was limited to virtual communication and meetings with parents which posed problems for those with connectivity or technological barriers Module refinement: The research team experienced significant difficulty in enabling the parent advisors to use Recollective software to beta-test that online modules due to connectivity and technology barriers

## Data Availability

No new data were created or analyzed in this study. Data sharing is not applicable to this article.

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
