# Peer review of "Lessons Learned with a Triad of Stakeholder Advisory Boards: Working with Adolescents, Mothers, and Clinicians to Design the TRUST Study"

_children, 2023, doi:10.3390/children10030483_

Round 1

Reviewer 1 Report

The study was robust and well-planned. Just a few comments for clarification. Kindly refer to the pdf version. Thank you and all the best. 

Reviewer 2 Report

REVIEW   

Lessons Learned with a Triad of Stakeholder Advisory Boards: 2 Working with Adolescents, Mothers, and Clinicians to Design 3 the TRUST Study.

Alexis Richards 1,†,* Marissa Raymond-Flesch 2,†, Shana D. Hughes 3, Yinglan Zhou 4 and Kimberly A. Koester

Analysing this work from the point of view of social research methodology, the following comments emerge:

1/ It is not really clear what the results of the research are and what the subject of the research consists of.

2/ The main criticism of the research is the methodological weakness of the sample design, sampling and implementation as well as the presentation of research details.

Although the selection of different types of respondents was planned and medical centres from different regions were selected [by ethnic group, rural and urban area], this translated poorly in the composition of the sample; thus, it was not possible to collect comparative data between the different types of respondents, apart from a comparison at a general level: medical professionals, parents  (only 3 mothers) and adolescents (e.g. there were only  (white?) students in the latter group). The authors themselves admit that they did not manage to reach all types of respondents.    Therefore, the saturation effect cannot appear.

The authors themselves point out that ethnic minority [ Hispanic] and rural residents are problematic groups when it comes to access to sexual health care. And at the same time these groups were not contacted. As is well known, the criterion of race is related to culture and wealth, which shapes attitudes towards sex, the level of traditionalism, the family model or, finally, the availability of health services.

Qualitative research is dedicated precisely to collecting information in the case of challenging respondents. Even in the case of difficulties, research is usually carried out until the desired result is obtained, 'to the point'.  One could even say that it is particularly the representatives of the Latinx population, the rural population that SHOULD be in the sample.

In the research design, it seems that a more optimal strategy for selecting respondents for the so-called triad would be to include related respondents who form real social networks, i.e. parents and their children and their doctors. It would provide information of the same processes from the perspective of different actors.

 Part of the description of the research process is very detailed (e.g. how contacts were made), but part of it is very laconic and scattered in different places (e.g. the information that Respondents are paid appears 'in passing', later in the article); the analyses would gain in clarity if the description of the methodology were put in order .  

Recruiting respondents for remuneration is important information and, in a situation where a respondent is paid to participate in the research, an analysis of the impact of this arrangement on the research results should be provided.

The authors address the issue of increasing the competence of the research team by creating a group of experts and using their knowledge of how to reach respondents.  Ironically, they themselves became victims of their inability to reach difficult respondents. And yet the purpose of this study was to find out how to reach difficult respondents.  They did not really explain why they failed to recruit difficult categories; in their explanations they refer to other studies: it is well known in social research that people of lower social status, marginalised  people are less socially active, have less trust in institutions, in science,

The issue of cultural factors has been neglected in the research; namely, sex is a taboo subject and it is cultural norms that shape the patterns of conversation on this topic.

Finally, it is worth adding that there is a lack of analysis of the work and the state of research in the field of social research methodology, while we already have extensive knowledge of how to conduct difficult research, the types of respondents, the difficulties of reaching respondents, dealing with difficult respondents and sensitive topics of conversation, and finally sampling methods, 

Table 2 - it is not clear whether the planned strategies, the three realised strategies are described in this table ? I guess it is about the realised strategy. There is no comment in the table about the effectiveness of the different methods of reaching the respondent (eg recruitment flyers posted in study clinical sites). If recruitment methods are analysed, the effectiveness of these methods should be described so that they can be compared somehow, e.g. a flyer vs telephone communication.

The descriptions of the respondent’s evaluation of the the TRUST intervention  are quite general, e.g. young people suggested to include images, but why and what types of images?  detailed analysis is lacking : was it a question of aesthetic, informational or technical criteria, or was it a question of the clarity of the message conveyed by the image; the comparison how the same element of TRUST were seen by different stakeholder groups.

This is important because it is on this conspecific information that the extensive knowledge on how to improve  the TRUST intervetnion and the techniques of reaching respondents in specific research problems and in a specific social context, e.g. the "social context", is based. 

I am not convinced that the following objective has been satisfactorily achieved:

“ [367] Our research team was committed to including three different stakeholder groups to optimize the effectiveness of the TRUST intervention and ensure representation  of all stakeholders in the study’s development, particularly adolescent voices. Overall, we  were able to engage the triad of stakeholders critical in adolescent healthcare, adolescents,  parents, and clinical advisors, independently and in parallel throughout multiple phases  in the TRUST study.”

Author Response

Reviewer 3: Analysing this work from the point of view of social research methodology, the following comments emerge:

1/ It is not really clear what the results of the research are and what the subject of the research consists of.

2/ The main criticism of the research is the methodological weakness of the sample design, sampling and implementation as well as the presentation of research details.

Response: We thank the reviewer for taking the time to provide feedback on our manuscript and regret that R2 was confused about the subject of our paper. The subject of this paper consists of reporting on the lessons learned from working with a set of stakeholders to develop an intervention, as is stated in the title. Because this paper is not a traditional reporting of results from a quantitative study nor the findings of a qualitative study, rather this is a methods paper reporting on the outcomes of our efforts to assemble a triad of community advisors, the main criticism of the “methodological weakness” does not apply. This paper does not intend to report on the TRUST Study research design, sampling frame or implementation of the study. It is a report of the work done during the developmental phase of the study. We have clarified this point in several sections of the manuscript.

Reviewer 3 Report

This is a well organized and carefully written account of TRUST = Technology-based Resources to Increase Uptake of Sexual Health Services for Teens which is an effort to form "integrated modules" among clinicians, adolescents and parents to facilitate access to medical information and services. It assumes that "...involvement of stakeholders in research can impact adolescent and pediatric outcomes as well as improve the engagement of people from underserved communities". TRUST appeared to involve a very large staff who performed a large number of activities. I would have liked to have known an approximate budget. Down the road, as this approach to forming advisory input gains more acceptance as I think will happen, some sort of cost-benefit accounting will have to investigated. 

Author Response

Reviewer 3: I would have liked to have known an approximate budget:

Response: We thank the reviewer for taking the time to provide feedback on our manuscript and study. The budget that we spent on recruitment and engagement of the three advisory boards was between $5000-$7500 dollars total. This includes $1500 for each site participating and $150-200 per advisor per year for participation depending on level on engagement.